# COVID-Related Distress Is Associated with Increased Menstrual Pain and Symptoms in Adult Women

**DOI:** 10.3390/ijerph20010774

**Published:** 2022-12-31

**Authors:** Laura A. Payne, Laura C. Seidman, Boyu Ren, Shelly F. Greenfield

**Affiliations:** 1McLean Hospital, Belmont, MA 02478, USA; 2Harvard Medical School, Boston, MA 02115, USA

**Keywords:** dysmenorrhea, COVID, stress, menstrual symptoms

## Abstract

The COVID-19 pandemic resulted in heightened stress for many individuals, with women reporting more stress than men. Although a large body of evidence has demonstrated that stress, in general, can impact the menstrual cycle, it is not yet clear if COVID-specific stress would impact women’s menstrual health. The current study explored the relationship between COVID-related stress and distress and menstrual variables (menstrual pain, number and severity of menstrual symptoms, and menstrual pain interference) in a sample of reproductive-age adult women. Seven-hundred fifteen women completed the initial survey and were re-contacted to complete the same survey three months later. Of those recontacted, 223 completed the follow-up survey. Results indicated that COVID-related stress and distress was associated with higher levels of menstrual pain, more frequent and more severe menstrual symptoms, and greater menstrual pain interference, even after accounting for age, hormonal use, bodily pain, and pain catastrophizing. Our findings suggest that women experience unique vulnerabilities that directly impact their health and functioning, and both research and clinical care should address these symptoms through careful assessment and treatment of menstrual pain and symptoms, particularly during and after periods of high stress and distress.

## 1. Introduction

The Coronavirus Infection Disease (COVID) pandemic dramatically changed the lives of most individuals in the world, beginning in February 2020. In an effort to curb the transmission and infection rates of the Severe Acute Respiratory Syndrome Coronavirus 2 (SARS-CoV-2), most communities in the United States and around the world implemented a series of “stay at home” orders, where public and private facilities, including schools and places of employment, and community gatherings were discouraged and/or prohibited. Anxiety about the virus itself, as well as the impact of the efforts to reduce the incidence of the virus, resulted in psychological stress for many individuals [1]. Women, in particular, were vulnerable to these stressors [2], in part due to their role as caregivers [3] and as essential workers [4]. These and other COVID-related stressors unique to women had a direct impact on their health and well-being [5].

A key indicator of women’s health is the functioning and cyclicity of the menstrual cycle [6,7], and there is a well-established link between psychological stress and menstrual cycle disruption via activation of the hypothalamic–pituitary–adrenal (HPA) and hypothalamic–pituitary–ovarian (HPO) axes [8]. Emerging reports have indicated disturbances in women’s menstrual cycles during the COVID pandemic, including changes in length of the menstrual cycle, duration of menses, and Premenstrual Syndrome (PMS) symptoms [9,10]. Specifically, those who reported higher levels of overall stress during the pandemic also reported greater menstrual symptoms, such as heavier bleeding during menstruation, longer menstrual cycles, and more severe symptoms of PMS. However, other studies have found different patterns, with the COVID pandemic associated with decreased duration of menstruation [11] or no changes in menstrual cycle length at all [12]. The lack of consistency in these findings may point to differences in assessment of stress, methodology (e.g., time frame that the study measures were administered in relation to the timing of the pandemic), or individual differences among women with regard to how stress impacts the menstrual cycle.

Menstrual pain (i.e., dysmenorrhea) is one common and disabling menstrual symptom for many reproductive-age women that may also be susceptible to stress experienced during the pandemic [13]. One study of Jordanian medical students reported significant increases in rates of severe dysmenorrhea and related activity disruption following the pandemic, although these changes were not linked to measures of stress, specifically [14]. A more recent cross-sectional study of 1335 women found that increased anxiety during the pandemic was associated with a change from non-painful to painful periods [15]. Additional data on the impact of COVID-related stress on non-menstrual pain populations supports the notion that increased stress during and due to the pandemic can worsen the pain symptom trajectory [16,17,18]. For example, adults with fibromyalgia reported higher levels of COVID-related fear and anxiety, compared to adults without fibromyalgia [19]. Worry and stress, as well as a pre-existing sensitivity to somatic symptoms, have been associated with self-report of greater somatic symptoms during the pandemic [20]. In a study of 150 patients with chronic pain, women were at greater risk of increased pain severity and pain interference, even after accounting for other demographic variables [21]. These findings all point to the importance of accounting for menstrual symptoms, menstrual pain, and overall body pain in a single study to better understand reproductive-age women’s responses to the COVID pandemic.

Given the impact of stressors and related distress on the menstrual cycle and pain conditions, the present study aimed to examine the relationship between COVID-related stress and distress and menstrual pain, menstrual symptoms, and menstrual pain interference in a large sample of reproductive-age women assessed during the pandemic and three months later. We hypothesized that COVID-related stress and distress would be associated with menstrual pain, menstrual symptom severity, number of menstrual symptoms, and menstrual pain interference, and these relationships would remain stable over the course of three months. Additionally, we hypothesized that these relationships would exist independent of other factors that may contribute to menstrual pain and symptoms, such as report of overall bodily pain, hormonal treatments used, and pain catastrophizing.

## 2. Materials and Methods

### 2.1. Participants

Participants for the current study were recruited from a convenience sample of individuals who had agreed to answer survey questions for a survey-based company (Market Cube, Inc., Schlesinger Group; see description below). Invitations were stratified by age to represent the age spread across the United States. We aimed to enroll participants to obtain a range of menstrual pain ratings, with 25% of participants reporting menstrual pain ratings of 0–2 on the 0–10 Numeric Rating Scale (NRS; see description under Measures), 60% reporting ratings of 3–7, and 15% reporting ratings of 8–10. Survey completion was monitored in real time and the REDCap screening page logic was modified periodically to temporarily close “bins” to ensure that approximately such a distribution was achieved. Inclusion criteria included: (1) female, aged 18–55 years; (2) at least one menstrual cycle during the previous 3 months; and (3) self-reported regular menstrual cycles during the previous 12 months. Participants could be either naturally menstruating or using 1 or more exogenous hormones. Exclusion criteria included: (1) not able to read and understand English; and (2) currently pregnant. 

Three thousand and thirty individuals completed the initial screening/eligibility questions in September 2020. Six participants were identified as duplicates of an existing record and removed. For three of these duplicates, the screening questions were identical both times and only one of the surveys was completed beyond the screening page. In these instances, the full survey was retained in the database, and the duplicate was removed. For one duplicate case, answers to the screening questions differed between the two cases, so both were removed. One additional case was removed because both the survey and the ineligibility page were completed, indicating that the participant paged back in the browser and changed her answers to the eligibility questions. 

One thousand three hundred and ninety-six individuals were excluded as ineligible due to pregnancy (*n* = 131), not having a period during the prior three months (*n* = 61), not having regular periods over the prior 12 months (*n* = 405), or neither having a period during the prior three months nor having regular periods over the prior 12 months (*n* = 799). An additional 336 individuals were excluded from completing the survey based on menstrual pain NRS bin being full. Of the 1292 who were eligible, 60 choose to not participate (i.e., they were directed to the survey but did not answer any questions). One thousand two hundred and thirty-two women at least partially completed the baseline survey and were then recontacted three months after the baseline survey (December 2020) for completion of the identical set of measures. For the current study, 715 participants had complete data from baseline and were included in the baseline analyses. Of these 715, 223 also completed the survey follow-up and were included in the baseline + follow-up group. See Figure 1 for a flow chart of study enrollment.

### 2.2. Procedures

Survey invitations were sent to women in the target age range who were registered as research panelists in UniVox (managed by Market Cube, Inc. (Mt Pleasant, SC, USA), Schlesinger Group (Iselin, NJ, USA)). Interested participants clicked the included link and were directed to four screening/eligibility questions assessing pregnancy, having a period over the prior three months, having regular periods over the prior 12 months, and average menstrual pain. Ineligible participants were redirected to a Market Cube webpage; eligible participants were directed to the information sheet and proceeded to the survey if they agreed to participate. Study data were collected and managed using REDCap (Research Electronic Data Capture [22,23]) electronic data capture tools hosted at McLean Hospital. REDCap is a secure, web-based software platform designed to support data capture for research studies, providing (1) an intuitive interface for validated data capture; (2) audit trails for tracking data manipulation and export procedures; (3) automated export procedures for seamless data downloads to common statistical packages; and (4) procedures for data integration and interoperability with external sources. For completing the survey, participants were compensated 200 points (equivalent to $2) within their UniVox account; points are redeemable for gift cards, etc. This study was reviewed by the Partners Healthcare IRB (protocol #2020P002578) and was determined to meet the criteria for exemption. Due to the anonymous nature of this survey study, informed consent was not required.

### 2.3. Measures

#### 2.3.1. Demographics

Demographic variables were obtained, including age, race, and education level.

#### 2.3.2. Menstrual History and Pain

For the purposes of this study, a self-report measure was included assessing menarche age, number of hormonal treatments currently using, whether the participant was menstruating in the last seven days (yes/no), average menstrual pain (without the use of pain medication) rated on a 0 (none) to 10 (worst pain imaginable) NRS [24,25]. An additional variable of interference in daily living due to menstrual pain (0; no interference to 10; complete interference NRS) was also included.

#### 2.3.3. Menstrual Symptoms

Endorsement and severity of menstrual symptoms beyond menstrual pain (e.g., low back pain, headache/migraine, nausea, diarrhea, etc.) was assessed with a well-established measure that was developed to evaluate and test symptom-based phenotypes of women with dysmenorrhea [26,27,28]. This measure asks participants to indicate the severity of each symptom on a 0 (not present) to 10 (extremely severe) scale over the past 6 months (when menstruating). For the purposes of this study, the total of the severity ratings for each of the symptoms endorsed (Menstrual Symptom Severity), and the number of menstrual symptoms endorsed (i.e., any symptom rated at a severity of at least “1”; Menstrual Symptom Count), were each used as outcome variables.

#### 2.3.4. Bodily Pain

Bodily pain was assessed using the Collaborative Health Outcomes Information Registry (CHOIR) Body Map [29], which is fully integrated into REDCap and allows each participant to select the body location where she has experienced pain the last month *while not menstruating*. The Body Map is a well-established measure of widespread pain [30] that can be used with chronic pain (e.g., [31]) and non-chronic pain populations [32,33,34]. For the current study, the total number of body map locations endorsed as being painful over the past month (range 0–74) was used. Additionally, participants rated body pain severity in the past month using a 0 (none) to 10 (worst pain imaginable) NRS.

#### 2.3.5. COVID Stress Scales (CSS)

The CSS [35] is a 36-item measure assessing stress and anxiety in response to the COVID-19 pandemic. The CSS is comprised of 5 sub-scales: danger and contamination fears (e.g., “I am worried that people around me will infect me with the virus”), fears about economic consequences (e.g., “I am worried about grocery stores running out of food”), xenophobia (e.g., “If I met a person from a foreign country, I’d be worried that they might have the virus”), compulsive checking and reassurance seeking (e.g., “Checked social media posts concerning COVID-19”), and traumatic stress symptoms (e.g., “I had trouble sleeping because I worried about the virus”) about COVID-19. 

#### 2.3.6. COVID-19 Exposure and Family Impact Survey—Adolescent and Young Adult Version (CEFIS-AYA)

The CEFIS-AYA [36,37] is a self-report measure that assesses participants’ experiences during the COVID-19 pandemic and the impact of the pandemic on psychological and social functioning. The Exposure subscale inquires whether the participant has been exposed to COVID and related events (e.g., “I had a stay at home order,” “I/we had difficulty getting food”) and consists of two parts (i.e., self and family member). Responses for this subscale were merged such that a yes responses in either subpart was scored as a yes response for the item. The Impact subscale consists of 15 items assessing the impact of the pandemic on the participant’s physical, emotional, and social life (e.g., relationships with friends, ability to be independent, substance use, etc.). Items are scored from 1 (“made it a lot better”) to 4 (“made it a lot worse”). An answer choice of “not applicable” was also available for each item. Impact subscale scores are calculated as the mean of all items as long as at least twelve of the subscale’s fifteen items were answered (i.e., not missing or answered as not applicable). Means < 2.5 indicate positive valence while means > 2.5 indicate negative valence. The Distress subscale is a single item numeric rating scale on which participants report the level of distress caused by the pandemic from 1 (no distress) to 10 (extreme distress). The AYA version of the measure was used as it includes additional questions that are relevant for the current study and is applicable to a broader range of participants than the original (i.e., caregiver only) version.

#### 2.3.7. Pain Catastrophizing Scale (PCS)

The PCS is a 13-item scale that asks participants to indicate the degree to which they have different thoughts when experiencing pain [38]. This measure assesses fears related to pain experiences. Although it includes three subscales (rumination, magnification, and helplessness), the total score on this measure was used for the analyses. 

### 2.4. Data Analysis

#### 2.4.1. Baseline Analysis

This analysis focuses on subjects with complete records of the relevant variables at either baseline only or at both baseline and follow-up. If a subject has complete records at both time points, she was included in the sensitivity analysis described below. Seven hundred and fifteen subjects were included in the final analysis. Linear regression was used to examine the association between the menstrual health variables (outcome measures) and variables derived from the COVID-related measures after adjusting for six covariates related to demographic and menstrual/health histories. The four menstrual health variables were used as the outcome variables in the regression models and analyzed separately. The reference level for race was “White,” and education was regarded as a continuous variable. T-test was used to assess the statistical significance of the regression coefficients. Since this study is conducted mainly for an exploratory purpose, no multiple testing correction was applied to the *p*-values.

#### 2.4.2. Sensitivity Analysis Using Baseline and Follow-Up Data

There were 223 subjects in the dataset with both baseline and follow-up measurements of the relevant variables. A sensitivity analysis was conducted on these subjects to test the robustness of the results from the baseline analysis. Since each subject now has two measurements for the time-varying variables, the generalized estimating equations (GEE) approach [39,40] was used to recover the population-level associations between the set of variables considered in the baseline analysis while accounting for the correlations between the repeated measures from the same subject. A compound symmetric correlation structure was assumed when fitting the GEE, and the *p*-value for a regression coefficient was obtained by using normal approximation on its robust z-score.

## 3. Results

Demographic characteristics for those in the baseline only and baseline + follow-up groups are shown in Table 1. There were no differences between those who completed the baseline only and those who completed the baseline + follow-up.

As shown in Table 2, for those who completed the baseline assessment, results indicated that CEFIS distress scores and body pain severity in the past month were positively related to average menstrual pain, while age, number of exogenous hormones currently using, and CEFIS impact scores were negatively related to average menstrual pain. Asian race tended to have lower level of pain compared to White when controlling for all other variables. A similar pattern was observed for those who completed both the baseline and follow-up assessments, with CSS danger and contamination, CEFIS distress, and body pain severity in the past month positively predicting average menstrual pain. Number of hormones currently using, and CEFIS impact scores negatively predicted average menstrual pain. Other races tended to have higher reported level of pain compared to White while Asian race still tended to have lower reported level of pain.

CSS compulsive checking, CEFIS exposure, CEFIS distress, body pain severity in the past month, and PCS scores positively predicted menstrual symptom severity, while Asian race had lower menstrual symptom severity compared to White in those completing the baseline assessment (Table 3). For those with both baseline and follow-up assessments, CSS compulsive checking, CEFIS exposure, CEFIS distress, and body pain severity in the past month positively predicted menstrual symptom severity.

CSS traumatic stress and menstrual pain in the past month were positively related to menstrual symptom count, and age and CSS danger and contamination were negatively related to menstrual symptom count in those who completed the baseline assessment only (Table 4). In participants with baseline and follow-up data, education level, CSS compulsive checking, and body pain severity in the past month were positively related to menstrual symptom count, and age was negatively related to menstrual symptom count.

When examining predictors of menstrual pain interference, CEFIS exposure, CEFIS distress, and body pain severity in the past month were positive predictors, and age and CEFIS impact were significant negative predictors in the baseline sample (Table 5). In the baseline and follow-up sample, CSS danger and contamination, CSS compulsive checking, CEFIS distress, and body pain severity in the past month emerged as positive predictors of menstrual pain interference, while CEFIS impact negatively predicted menstrual pain interference. Other races were associated with a lower level of pain interference compared to White.

## 4. Discussion

The current study aimed to clarify the relationship of COVID-related stress and distress, menstrual pain and related symptoms, and menstrual pain interference, after accounting for demographic and menstrual variables, self-reported (non-menstrual) bodily pain, and pain catastrophizing in a large sample of reproductive-age women. Given the well-established link between stress and menstrual cycle changes, as well as more recent evidence linking COVID-specific stress to menstrual changes and, separately, to increased pain and somatic symptoms, we hypothesized that COVID-related stress and distress would both be associated with increased menstrual pain, number and severity of menstrual symptoms, and menstrual pain interference.

Data analyses provided support for this hypothesis, with many measures positively predicting the four outcome variables, which suggests that higher levels of COVID stress and distress were associated with higher levels of menstrual pain and symptoms. In particular, CEFIS distress (a single item asking participants how much distress they experienced as a result of the pandemic) was positively related to all outcome measures, with the exception of menstrual symptom count, in both the baseline only and baseline + follow-up samples. On the other hand, CEFIS exposure (a subscale that quantifies the number of COVID-related exposures to stressful situations) was not a significant predictor of the menstrual outcome variables (with the exception of Menstrual Symptom Severity). These data suggest that the experience of distress, regardless of the actual number of stressful events, may have the greatest impact on menstrual pain and related symptoms.

Additionally, body pain severity showed a strong, positive relationship with all four menstrual-related outcome variables. This is consistent with previous data in clinical populations suggesting that many women who experience chronic pain also experience menstrual pain [41,42,43]. Although our data did not identify those with chronic pain, the results still suggest a strong link between overall body pain severity and menstrual pain, symptoms, and interference. However, COVID-related stress and distress variables were still significantly related to the menstrual outcome variables over and above the relationship with body pain.

Interestingly, Asian race was negatively associated with most outcome variables. Little work has been done on racial/ethnic differences in dysmenorrhea, particularly among Asian women. However, research in Asian populations have reported a general reluctance to seek help for menstrual-related problems [44,45], which may also reflect a tendency to view symptoms of dysmenorrhea as “normal,” and perhaps result in lower overall pain scores. Studies examining racial and ethnic differences in the prevalence of self-reported chronic pain have also found lower prevalence in Asian populations [46,47], although other research suggests *heightened* pain sensitivity in response to laboratory pain tasks in Asian individuals [48,49,50]. The data from the current study supports this notion that the relationship between racial differences and pain is complex, involving many social and cultural factors.

Another interesting finding was that the number of hormones a participant identified as using was *negatively* associated with menstrual pain in both the baseline and baseline + follow-up samples. Hormonal interventions are commonly used for menstrual pain, although in this study we required individuals to still have regular menstrual cycles so they could not be fully suppressing the menstrual cycle. The negative relationship with hormonal use and menstrual pain suggests that reduction of pain is at least somewhat effective, even without menstrual suppression.

Results of this study demonstrated a strong relationship between COVID stress and distress and menstrual pain, initially assessed in the first six months of the pandemic and prospectively three months later. These findings are consistent with the one other study reporting increased rates of dysmenorrhea following the pandemic [14]. Yet, the pathophysiological mechanisms by which pandemic-related stress and distress affect menstrual pain are not entirely clear. Stress is associated with increased synthesis of uterine prostaglandins [51,52,53], which is a known factor contributing to menstrual pain in many women. However, alterations in pain processing via the central nervous system may also be affected by stress [54] and result in heightened menstrual pain [55,56]. Future research examining the contribution of each of these variables is the important next step to identifying unique risk factors of menstrual pain and symptoms in women.

### Limitations

The current study has a number of limitations that warrant discussion. First, we did not obtain data on participants’ menstrual pain, menstrual symptoms, or menstrual pain interference prior to the onset of the pandemic. Therefore, we cannot conclude that the pandemic or pandemic-related stress and distress *caused* any changes; rather, we are able to determine only that there is a relationship between stress and distress about the pandemic and menstrual pain and symptoms. Additionally, our findings suggest that these relationships were stable, at least over a three-month period. We do not anticipate any bias that would affect completion of the follow-up survey, so these findings appear consistent. Third, we included two measures of COVID-related stress and distress to try to capture various aspects of experiences that people may be having as a result of the pandemic. However, these two measures provided only a limited picture of the many stressors that women may have experienced, so it is important to recognize that we cannot fully understand the total impact of the stress of the pandemic based on these data. Another important consideration is that we did not obtain any vaccine-related data, including vaccination status, timing of the last vaccine, or stress related to the vaccine, specifically. Given emerging evidence of the impact of the COVID vaccine on the menstrual cycle [57,58,59], it is possible that vaccination status may have impacted our findings. However, given that we saw similar patterns at baseline and 3-month follow-up, vaccination status may have had less of an impact, at least on the menstrual variables we included in this study. Additionally, we asked about *types* of exogenous hormone use (e.g., “pill,” IUD, hormonal patch), but we did not obtain any more detailed information about the use of hormones (e.g., for how long or what type of pill).

## 5. Conclusions

These data show that women who experienced stress related to the COVID pandemic also experienced higher levels of menstrual pain, more frequent and more severe menstrual symptoms, and more menstrual pain interference. These relationships were true, even after accounting for age, the number of exogenous hormones being used, bodily pain, and pain catastrophizing. Additionally, the number of COVID-related stressors experienced was not strongly associated with outcome variables. Our findings suggest that women experience unique vulnerabilities that directly impact their menstrual health and overall functioning, and both research and clinical care should address health through careful assessment and treatment of menstrual pain and symptoms, particularly during and after periods of high stress and distress. Clinical interventions focused on reducing stress may be particularly effective for helping women with menstrual symptoms, even if the stressors they are experiencing cannot be changed.

## Figures and Tables

**Figure 1 ijerph-20-00774-f001:**
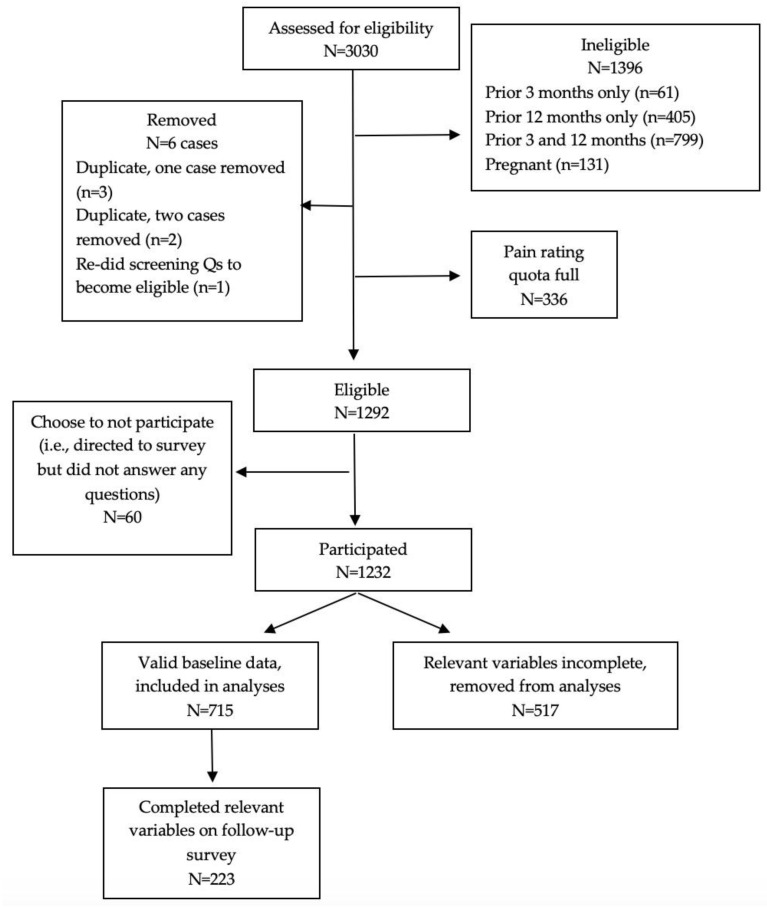
Enrollment flow diagram.

**Table 1 ijerph-20-00774-t001:** Descriptive statistics of demographic and clinical variables at baseline for the two groups of subjects (Baseline only vs. Baseline + follow-up).

	Baseline Only (*n* = 715)	Baseline + Follow-Up (*n* = 223)
**Age**		
Mean (SD)	34.5 (9.19)	37.4 (8.42)
**Education**		
Some high school	25 (3.5%)	3 (1.3%)
High school diploma or GED	149 (20.8%)	43 (19.3%)
Some college or 2-year degree	202 (28.3%)	48 (21.5%)
4-year college graduate	211 (29.5%)	86 (38.6%)
Some school beyond college	14 (2.0%)	3 (1.3%)
Graduate or professional degree	114 (15.9%)	40 (17.9%)
**Race**		
White	546 (76.4%)	172 (77.1%)
Black or African American	66 (9.2%)	18 (8.1%)
Asian	74 (10.3%)	27 (12.1%)
American Indian/Native Hawaiian/Multi-Racial	29 (4.1%)	6 (2.7%)
**Number of Hormones using**		
Mean (SD)	0.425 (0.691)	0.455 (0.734)
**Age at menarche**		
Mean (SD)	12.3 (1.76)	12.6 (1.78)
**Had period in the past 7 days (y/n)**		
Percent indicating “yes”	46.7%	45.2%

**Table 2 ijerph-20-00774-t002:** Predictors of Average Menstrual Pain in participants who completed the baseline assessment only and those who completed both baseline and follow-up assessment.

	Baseline Only (*n* = 715)			Baseline + Follow-Up (*n* = 223)		
Predictors	*Estimates*	*CI*	*p*	*Estimates*	*CI*	*p*
(intercept)	5.39	3.90–6.88	**<0.001**	3.80	0.67–6.93	**0.017**
Age	−0.02	−0.04–−0.0	**0.021**	−0.01	−0.05–0.03	0.526
Black or African American race *	−0.05	−0.60–0.50	0.854	−0.98	−2.41–0.44	0.176
Asian race *	−0.68	−1.21–−0.15	**0.011**	−1.13	−2.18–−0.08	**0.036**
American Indian/Native Hawaiian/Multi-Racial *	0.45	−0.35–1.24	0.270	2.65	0.65–4.64	**0.009**
Education level	−0.12	−0.24–−0.0	0.06	0.09	−0.16–0.34	0.481
Number of Hormones using	−0.34	−0.60–−0.09	**0.008**	−0.51	−1.01–−0.02	**0.043**
Age at menarche	−0.03	−0.12–0.06	0.511	−0.03	−0.20–0.15	0.760
Period in last 7 days	−0.12	−0.42–0.17	0.417	−0.08	−0.60–0.44	0.755
CSS_dc	0.01	−0.00–0.03	0.130	0.06	0.02–0.10	**0.001**
CSS_s	−0.01	−0.04–0.02	0.620	−0.01	−0.07–0.06	0.842
CSS_x	0.00	−0.03–0.05	0.899	−0.02	−0.08–0.05	0.613
CSS_t	0.01	−0.03–0.05	0.677	−0.01	−0.08–0.06	0.829
CSS_ch	−0.02	−0.05–0.02	0.443	0.02	−0.06–0.10	0.579
CEFIS (part 1; exposure)	0.01	−0.03–0.05	0.645	−0.01	−0.07–0.06	0.806
CEFIS (part 2; impact)	−0.24	−0.43–−0.05	**0.014**	−0.26	−0.51–−0.01	**0.045**
CEFIS (part 2; distress)	0.20	0.12–0.28	**<0.001**	0.20	0.06–0.35	**0.006**
Body pain in the past month	0.28	0.22–0.34	**<0.001**	0.20	0.10–0.31	**<0.001**
Body map # of locations	0.00	−0.02–0.03	0.701	−0.00	−0.05–0.04	0.865
PCS	0.01	−0.01–0.02	0.340	−0.01	−0.04–0.02	0.524

*Note.* * racial group is in comparison to White (reference group). **Bold** indicates statistical significance of *p* < 0.05. CSS = COVID Stress Scales; dc = danger and contamination; s = socio-economic; x = xenophobia; t = traumatic stress; ch = compulsive checking; CEFIS = COVID-19 Exposure and Family Impact Scale; PCS = Pain Catastrophizing Scale.

**Table 3 ijerph-20-00774-t003:** Predictors of Menstrual Symptom Severity in participants who completed the baseline assessment only and those who completed both baseline and follow-up assessment.

	Baseline Only (*n* = 715)			Baseline + Follow-Up (*n* = 223)		
Predictors	*Estimates*	*CI*	*p*	*Estimates*	*CI*	*p*
(intercept)	26.78	10.91–42.65	**<0.001**	16.35	−17.51–50.20	0.344
Age	−0.12	−0.31–0.07	0.222	−0.37	−0.80–0.07	0.101
Black or African American race *	3.88	−1.97–9.73	0.194	8.39	−7.02–23.80	0.286
Asian race *	−7.81	−13.44–−2.17	**0.007**	−5.84	−17.24–5.56	0.315
American Indian/Native Hawaiian/Multi-Racial *	−0.31	−8.80–8.18	0.943	15.99	−5.66–37.65	0.148
Education level	−0.29	−1.57–0.99	0.659	2.03	−0.66–4.72	0.138
Number of Hormones using	1.16	−1.54–3.87	0.399	2.23	−3.14–7.59	0.416
Age at menarche	−0.36	−1.32–0.59	0.457	−0.96	−2.84–0.93	0.321
Period in last 7 days	1.10	−2.07–4.26	0.497	2.02	−3.47–7.52	0.471
CSS_dc	−0.12	−0.31–0.08	0.250	0.22	−0.16–0.60	0.258
CSS_s	0.12	−0.23–0.48	0.497	0.15	−0.55–0.84	0.678
CSS_x	0.18	−0.16–0.53	0.289	0.09	−0.58–0.75	0.801
CSS_t	0.30	−0.11–0.70	0.148	−0.17	−0.95–0.60	0.664
CSS_ch	0.47	0.06–0.89	**0.024**	1.04	0.20–1.88	**0.015**
CEFIS (part 1; exposure)	0.41	0.01–0.81	**0.042**	0.94	0.24–1.64	**0.009**
CEFIS (part 2; impact)	−1.72	−3.73–0.28	0.092	0.66	−2.01–3.32	0.630
CEFIS (part 2; distress)	1.23	0.37–2.09	**0.005**	1.66	0.11–3.22	**0.036**
Body pain in the past month	4.20	3.57–4.82	**<0.001**	3.73	2.61–4.85	**<0.001**
Body map # of locations	0.15	−0.10–0.41	0.230	0.07	−0.39–0.53	0.767
PCS	0.21	0.05–0.37	**0.011**	−0.10	−0.43–0.23	0.539

*Note.* * racial group is in comparison to White (reference group). **Bold** indicates statistical significance of *p* < 0.05. CSS = COVID Stress Scales; dc = danger and contamination; s = socio-economic; x = xenophobia; t = traumatic stress; ch = compulsive checking; CEFIS = COVID-19 Exposure and Family Impact Scale; PCS = Pain Catastrophizing Scale.

**Table 4 ijerph-20-00774-t004:** Predictors of Menstrual Symptom Count in participants who completed the baseline assessment only and those who completed both baseline and follow-up assessment.

	Baseline Only (*n* = 715)			Baseline + Follow-Up (*n* = 223)		
Predictors	*Estimates*	*CI*	*p*	*Estimates*	*CI*	*p*
(intercept)	8.93	6.84–11.02	**<0.001**	9.17	4.85–13.48	**<0.001**
Age	−0.03	**−0.06–−0.01**	**0.008**	−0.07	−0.13–−0.02	**0.010**
Black or African American race *	0.10	−0.68–0.87	0.808	1.26	−0.071–3.22	0.210
Asian race *	−0.49	−1.24–0.26	0.199	−0.44	−1.90–1.02	0.553
American Indian/Native Hawaiian/Multi-Racial *	0.19	−0.93–1.32	0.734	1.36	−1.41–4.13	0.337
Education level	0.10	−0.07–0.27	0.232	0.44	0.10–0.79	**0.011**
Number of Hormones using	0.23	−0.13–0.59	0.210	0.02	−0.66–0.70	0.953
Age at menarche	−0.03	−0.16–0.09	0.617	−0.12	−0.36–0.12	0.327
Period in last 7 days	0.12	−0.29–0.53	0.552	0.33	−0.35–1.02	0.340
CSS_dc	−0.03	−0.05–0.00	**0.047**	0.00	−0.05–0.05	0.960
CSS_s	0.03	−0.02–0.07	0.274	0.00	−0.08–0.09	0.947
CSS_x	0.03	−0.02–0.07	0.265	0.04	−0.04–0.13	0.327
CSS_t	0.07	0.02–0.12	**0.007**	0.06	−0.03–0.16	0.199
CSS_ch	0.05	−0.00–0.11	0.051	0.11	0.00–0.21	**0.044**
CEFIS (part 1; exposure)	−0.00	−0.05–0.05	0.977	0.05	−0.04–0.13	0.311
CEFIS (part 2; impact)	−0.18	−0.44–0.07	0.157	−0.06	−0.38–0.27	0.736
CEFIS (part 2; distress)	−0.01	−0.13–0.10	0.798	−0.07	−0.26–0.13	0.507
Body pain in the past month	0.33	0.25–0.41	**<0.001**	0.24	0.10–0.38	**0.001**
Body map # of locations	0.02	−0.02–0.05	0.287	0.03	−0.03–0.08	0.368
PCS	0.01	−0.01–0.03	0.225	−0.02	−0.06–0.03	0.453

*Note.* * racial group is in comparison to White (reference group). **Bold** indicates statistical significance of *p* < 0.05. CSS = Covid Stress Scales; dc = danger and contamination; s = socio-economic; x = xenophobia; t = traumatic stress; ch = compulsive checking; CEFIS = COVID-19 Exposure and Family Impact Scale; PCS = Pain Catastrophizing Scale.

**Table 5 ijerph-20-00774-t005:** Predictors of Menstrual Pain Interference in participants who completed the baseline assessment only and those who completed both baseline and follow-up assessment.

	Baseline Only (*n* = 715)			Baseline + Follow-Up (*n* = 223)		
Predictors	*Estimates*	*CI*	*p*	*Estimates*	*CI*	*p*
(intercept)	3.75	2.13–5.38	**<0.001**	2.67	−0.58–5.92	0.107
Age	−0.03	−0.05–−0.01	**0.004**	−0.02	−0.06–−0.02	0.372
Black or African American race *	0.49	−0.11–1.09	0.107	0.20	−1.27–1.68	0.788
Asian race *	−0.30	−0.88–0.28	0.308	−0.69	−1.78–0.39	0.209
American Indian/Native Hawaiian/Multi-Racial *	0.40	−0.47–1.27	0.364	2.22	0.16–4.27	**0.034**
Education level	0.01	−0.13–0.14	0.939	0.20	−0.06–0.46	0.130
Number of Hormones using	−0.12	−0.40–0.15	0.381	−0.20	−0.71–0.32	0.456
Age at menarche	−0.03	−0.13–0.07	0.552	−0.10	−0.28–0.08	0.289
Period in last 7 days	−0.13	−0.45–0.20	0.450	0.08	−0.47–0.64	0.771
CSS_dc	0.01	−0.01–0.03	0.249	0.08	0.04–0.11	**<0.001**
CSS_s	0.01	−0.03–0.04	0.671	−0.01	−0.08–0.05	0.688
CSS_x	−0.02	−0.06–0.01	0.258	−0.06	−0.12–0.01	0.077
CSS_t	0.02	−0.02–0.06	0.262	−0.03	−0.11–0.05	0.425
CSS_ch	0.02	−0.02–0.06	0.357	0.10	0.01–0.18	**0.027**
CEFIS (part 1; exposure)	0.05	0.00–0.09	**0.030**	0.03	−0.04–0.10	0.362
CEFIS (part 2; impact)	−0.30	−0.51–−0.09	**0.005**	−0.29	−0.56–−0.01	**0.041**
CEFIS (part 2; distress)	0.19	0.10–0.27	**<0.001**	0.20	0.04–0.35	**0.013**
Body pain in the past month	0.30	0.24–0.36	**<0.001**	0.25	0.14–0.36	**<0.001**
Body map # of locations	0.00	−0.02–0.03	0.716	−0.00	−0.05–0.04	0.897
PCS	0.01	−0.00–0.03	0.153	−0.00	−0.03–0.03	0.979

*Note.* * racial group is in comparison to White (reference group). **Bold** indicates statistical significance of *p* < 0.05. CSS = COVID Stress Scales; dc = danger and contamination; s = socio-economic; x = xenophobia; t = traumatic stress; ch = compulsive checking; CEFIS = COVID-19 Exposure and Family Impact Scale; PCS = Pain Catastrophizing Scale.

## Data Availability

The current approved institutional mechanism for data sharing is by individual data use agreements executed between the interested parties.

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
