# Peer review of "COVID-Related Distress Is Associated with Increased Menstrual Pain and Symptoms in Adult Women"

_ijerph, 2022, doi:10.3390/ijerph20010774_

Round 1
Reviewer 1 Report
What is meant by "hormonal use" ? Adequate doses of a progestin and certainly an estrogen-progestin combination should be able to either1) abolish menstruation with or without ovulation inhibition or ii) minimize bleeding and pain by its anti-inflammatory effect.
The exacerbation of menstrual symptoms in conditions of stress is further argument that currently menstruation is the abnormal and not the default state(pregnancy) and it should be ameliorated by hormonal therapy. see
"Modern menstruation: Is it abnormal and unhealthy?" Medical Hypotheses. 2020;144:109955. 4.
Author Response
We very much appreciate Reviewer 1’s comment requesting clarification of the term “hormonal use,” which was used in the Conclusion section. We have changed that terminology to “the number of exogenous hormones being used” to be consistent with other sections of the manuscript. Additionally, our limited assessment of use of exogenous hormones (including specific pill combinations) was included as a limitation in the Discussion.
Reviewer 2 Report
This was a very insightful article to read and review and the authors should be commended for the work they have completed. The work on menstrual cycle status and health in premenopausal females during the COVID-19 pandemic is a highly relevant topic. Insights into probable contributors to menstrual cycle variations as a result of the stress experienced in the pandemic provide the foundation for future research and policies.
I have only minor considerations for the author to consider. Within the methods section, it was unclear to me if the eligible participants were required to be naturally menstruating or a combination of hormonal contraceptive users and naturally menstruating. Subsequently, in the results, I noted that there were negative correlations in females who had stated their use of hormonal contraceptives. This result would be expected due to the effect of hormonal contraceptives on HPO axis suppression and documented use for mitigating menstrual cycle symptoms and adverse disorders (e.g. endometriosis). I noted that this result was not elaborated on in the discussion and I felt that this is a result that the authors may consider clarifying in their write-up (both methods and discussion).
With some research suggesting vaccines for COVID-19 had impacts on cycle presentation and symptoms (variables assessed in this study), was vaccination status, the timing of the last vaccine, or stress related to COVID vaccine considered as a factor within this analysis? If not could this be considered in the discussion and or limitations section of the article?
A minor amendment to line 205: could the authors consider correcting to 'seven hundred and fifteen'
Within the tables are the authors able to provide a footnote for the # symbol that is presented in the hormone use row please?
Author Response
Reviewer 2 brought up some very interesting and helpful feedback that has strengthened the manuscript. We have now clarified that the sample included both naturally menstruating individuals as well as those using 1 or more exogenous hormones. We also expanded upon the finding that this reviewer highlighted – that hormonal use was negatively associated with menstrual pain.
This reviewer also raised a very important point about the potential impact of the COVID vaccine on the menstrual cycle for participants in this study. We did not obtain vaccine-related information for this cohort, and we have described this as a limitation in the manuscript. However, given that we saw similar patterns at baseline and 3-month follow-up, vaccination status may have had less of an impact, at least on the menstrual variables we included in this study.
We have corrected the number to “seven hundred and fifteen.”
The “#” symbol is only reflective of the word “#”.